# Exploring the Prevalence and Risk Factors of MASLD in Patients with Newly Diagnosed Diabetes Mellitus: A Comprehensive Investigation

**DOI:** 10.3390/jcm14103513

**Published:** 2025-05-17

**Authors:** Hatice Beyazal Polat, Mehmet Beyazal, Medeni Arpa, Bayram Kızılkaya, Teslime Ayaz, Ömer Lütfi Gündoğdu, Kamil Konur, Zehra Polat, Fatma Beyazal Çeliker, Halil Atasoy

**Affiliations:** 1Recep Tayyip Erdoğan University Faculty of Medicine, 53100 Rize, Türkiye; mehmet.beyazal@erdogan.edu.tr (M.B.); medeni.arpa@erdogan.edu.tr (M.A.); dr.bayramkizilkaya@hotmail.com (B.K.); teslime.ayaz@erdogan.edu.tr (T.A.); omerlutfi.gundogdu@erdogan.edu.tr (Ö.L.G.); kamilkonur@gmail.com (K.K.); fabeceliker@gmail.com (F.B.Ç.); halil.rakici@erdogan.edu.tr (H.A.); 2Rize Chambers and Commodity Exchanges Union Science High School, 53020 Rize, Türkiye; zehra.polat2104@gmail.com

**Keywords:** type 2 diabetes mellitus, metabolic dysfunction-associated steatotic liver disease, predictors

## Abstract

**Background:** Metabolic Dysfunction-Associated Steatotic Liver Disease (MASLD) represents a growing concern in the context of metabolic disorders, particularly among individuals diagnosed with type 2 diabetes mellitus (T2DM). This study aimed to investigate the prevalence of MASLD among newly diagnosed T2DM patients and identify the risk factors for MASLD in this population. **Methods**: This prospective study included 128 patients with newly diagnosed T2DM between January 2022 and June 2023. Demographic, clinical, anthropometric (BMI, waist circumference), and laboratory data (glucose, HbA1c, lipid profile, ALT, AST, creatinine, platelet count) were collected. MASLD was diagnosed based on ultrasonographic evidence of hepatic steatosis with at least one cardiometabolic risk factor after excluding other causes. Linear regression models were used to determine independent predictors. **Results**: MASLD was detected in 80.4% of patients. Compared with the MASLD (−) group, the MASLD (+) group had significantly higher ALT (47.1 ± 23 U/L vs. 24.9 ± 8 U/L, *p* < 0.001) and non-HDL cholesterol (189 ± 57 mg/dL vs. 167 ± 28 mg/dL, *p* = 0.047). Spearman correlation showed positive associations of MASLD severity with waist circumference, LDL cholesterol, and platelet count. ALT and BMI were independently associated with MASLD in linear regression analysis. **Conclusions:** This study underscores the significant prevalence of MASLD in newly diagnosed T2DM patients, emphasizing the relevance of early detection in addressing this common comorbidity in the diabetic population.

## 1. Introduction

Type 2 diabetes mellitus (T2DM) is a prevalent and complex metabolic disorder associated with various complications, which significantly impact public health [1]. Among the comorbidities linked with T2DM, Metabolic Dysfunction-Associated Steatotic Liver Disease (MASLD) has emerged as a noteworthy condition [2]. MASLD is characterized by hepatic steatosis in the presence of cardiometabolic risk factors and, by definition, it excludes other causes of liver fat accumulation such as excessive alcohol consumption or viral hepatitis [3,4]. MASLD is the most prevalent chronic liver condition and ranks among the leading causes of severe liver disease worldwide. The acknowledgment of MASLD as the predominant cause of chronic liver disease has been steadily growing, with a prevalence reaching up to 30% [5,6].

MASLD, which occurs as part of metabolic syndrome, is strongly associated with T2DM, obesity, and cardiovascular diseases. MASLD can progress to serious liver disease and hepatic complications [7]. It may also contribute to systemic inflammation and overall metabolic health problems [8]. Therefore, early recognition of MASLD in newly diagnosed T2DM patients is important for the appropriate management and development of targeted interventions. Recent studies have emphasized the importance of timely lifestyle and pharmacological interventions to halt or reverse disease progression in patients with MASLD, especially in its early stages [9]. This study uniquely investigates the prevalence and independent predictors of MASLD in a cohort of newly diagnosed T2DM patients, a population in which MASLD is frequently underdiagnosed at an early stage.

## 2. Methods

This prospective study included a total of 220 patients who were newly diagnosed with T2DM between January 2022 and June 2023. Among them, 128 patients met the inclusion criteria and were enrolled in the study. Fatty liver was observed in 103 patients. Each patient’s diagnosis of T2DM was confirmed by an internal medicine specialist using the diagnostic criteria of the American Diabetes Association [10]. Patients with type 1 DM, Wilson disease, HBV/HCV infections, a history of severe alcohol consumption, malignancies, severe liver/kidney insufficiency, and diabetes due to corticosteroid use were excluded from the study. Patients with a history of significant alcohol consumption (defined as >30 g/day for men and >20 g/day for women) were excluded. The participants were informed about the purpose and content of the study, and their consent was obtained before the commencement of the examinations (Figure 1). All newly diagnosed T2DM patients during the study period were considered for inclusion. Patients who declined participation (n = 8) were excluded. No randomization was performed. These exclusion criteria were applied to avoid confounding causes of hepatic steatosis.

The baseline information of patients, namely age, sex, medical history, clinical manifestations, laboratory values, and demographic details, was retrieved from the hospital’s electronic medical records. The patient’s body weight, height, waist circumference, and body mass index (BMI) were measured. All anthropometric measurements were performed by trained staff using calibrated instruments and following standardized procedures. Venous blood samples were collected from all patients after an 8 h fasting period for the following biochemical analyses: hemogram, glucose, creatinine, glycosylated hemoglobin (HbA1c), aspartate transferase (AST), alanine transaminase (ALT), and lipid panel. ALT was included due to its role in hepatocellular injury, while lipid parameters such as LDL and non-HDL cholesterol were assessed for metabolic risk.

### 2.1. DefinitionsandCalculations

Body mass index (BMI) was calculated by dividing body weight (kg) by the square of height (m^2^). The diagnosis of T2DM was based on the following diagnostic criteria outlined by the American Diabetes Association (ADA): (i) fasting plasma glucose level ≥ 126 mg/dL, (ii) 2 h plasma glucose level ≥ 200 mg/dL during a 75 g oral glucose tolerance test, (iii) random plasma glucose ≥ 200 mg/dL in a patient exhibiting classic symptoms of hyperglycemia or hyperglycemic crisis, (iv) hemoglobin A1c (HbA1c) level ≥ 6.5 [11]. The diagnosis of MASLD was made based on ultrasonographic evidence of hepatic steatosis in the presence of at least one cardiometabolic risk factor (CMRF), after excluding other known causes of steatosis [12]. The grading of MASLD based on liver USG is as follows: grade 1—mild diffuse increase in echogenicity is observed; grade 2—moderate increase in echogenicity, accompanied by a reduction in the visibility of the portal vein wall and diaphragm; and grade 3—increase in echogenicity, leading to the invisibility of the portal vein wall, diaphragm, and posterior part of the liver [13].

The presence of microvascular complications such as retinopathy, neuropathy, and nephropathy in patients with newly diagnosed T2DM was investigated. The presence of diabetic retinopathy was determined by fundus examination, performed by an ophthalmologist using a biomicroscope with a 66-gauge lens [14]. The presence of diabetic neuropathy was determined by neurological examinations and electromyography (EMG), performed by a neurologist. Median, ulnar, radial, peroneal, tibial, sural, and superficial motor and sensory nerve transmissions were examined in usually four or at least three extremities. Recording electrodes [15] were used to record motor signals and ring electrodes were used to record sensory nerve signals that were transmitted antidromically [16]. Persistent albuminuria on two or more occasions, with at least three months between each assessment, based on early morning urine samples, indicated the presence of diabetic nephropathy. Hypertension was characterized by consistent blood pressure readings ≥ 140/90 mmHg on at least three separate occasions or a self-reported diagnosis of hypertension. The diagnosis of coronary artery disease (CAD) was based on angiography measurements of coronary artery stenosis of at least 50% in one or more coronary arteries on an angiography machine [17]. The carotid artery intima-media thickness was determined using non-invasive USG.

This study was conducted in accordance with the Declaration of Helsinki (ethical approval no: E-40465587-050.01.04-716, 2023/131). Informed consent was obtained from all participants. The study protocol was approved by the Ethics Committee of Recep Tayyip Erdoğan University (approval no: E-40465587-050.01.04-716).

### 2.2. StatisticalAnalysis

All statistical analyses were performed using the Statistical Package for the Social Sciences 25.0 for Windows (SPSS Inc., Chicago, IL, USA). The normality of data was assessed using the Kolmogorov–Smirnov test. Continuous data were presented as mean ± SD, and for parameters with non-normal distribution, the median (25–75) percentiles were reported. Categorical data were expressed as percentages. The Chi-square test was employed to evaluate differences in categorical variables between the groups. For comparing unpaired samples, either Student’s *t*-test or Mann–Whitney U test was applied as appropriate. Spearman’s correlation analysis was employed to assess the relationships between fatty liver grades and relevant variables. The correlation coefficients (r) and *p*-values were reported. Spearman’s correlation coefficient was used for variables that were not normally distributed or ordinal in nature. Additionally, multiple linear regression analyses using the stepwise method were performed to assess the independent variables affecting the dependent variable MASLD. All independent variables in the multiple linear regression were tested for multicollinearity. If the variance inflation factor (VIF) exceeded 3.0, the variable was considered to be collinear. All reported confidence interval (CI) values are calculated at the 95% level. The significance level for all tests was set at *p* < 0.05.

## 3. Results

The demographic, clinical, and laboratory characteristics of patients with and without hepatic steatosis are presented in Table 1. Of the 128 patients included in the study, 103 (80.4%) were diagnosed with fatty liver, while 25 (17.6%) had no evidence of steatosis. The distribution of age and sex was comparable between patients with and without fatty liver. Similarly, no notable differences were identified in clinical features such as BMI, waist circumference, retinopathy, nephropathy, neuropathy, coronary artery disease, hypertension, or smoking status across the two groups.

Regarding biochemical parameters, glucose, HbA1c, LDL cholesterol, triglyceride, HDL cholesterol, AST, creatinine levels, and platelet count showed similar values between the groups. In contrast, ALT and non-HDL cholesterol levels were significantly higher in patients with fatty liver (*p* < 0.005) (Table 1).

The relationship of fatty liver grades with other variables using Spearman’s correlation analysis is presented in Table 2. While fatty liver was not detected in 18% of the patients, grade I was detected in 13%, grade II was detected in 58%, and grade III was detected in 12%. Spearman’s correlation analysis demonstrated a positive correlation between MASLD of all grades and waist circumference (r = 0.234, *p* = 0.032), LDL cholesterol (r = 0.263, *p* = 0.021), and platelet count (r = 0.226, *p* = 0.040) (Table 2).

LDL cholesterol levels increased progressively with the severity of MASLD. The mean LDL levels were 137.2 ± 14.7 mg/dL in grade 1, 153.3 ± 18.5 mg/dL in grade 2, and 183.7 ± 21.5 mg/dL in grade 3 (*p* = 0.041) (Figure 2).

The MASLD parameters and statistically significant variables were included in the regression analysis. In the regression analysis of the factors associated with MASLD, ALT and BMI were found to be independent predictors of MASLD; cIMT, waist circumference, glucose, HbA1c, LDL cholesterol, triglyceride, non-HDL, AST, creatinine, platelet count, and HDL cholesterol did not independently predict MASLD (Table 3).

In the comparison of ultrasonographic findings with LDL cholesterol levels, the severity of steatosis was correlated with LDL cholesterol levels (Figure 3).

## 4. Discussions

In this study, we aimed to define the prevalence of MASLD and associated risk factors in T2DM patients. We found that (i) ALT levels and non-HDL cholesterol were elevated in the fatty liver group. (ii) Spearman’s correlation analysis revealed a positive association between the grades of fatty liver and waist circumference, LDL cholesterol levels, and platelet count.

(iii) Regression analysis revealed that ALT and BMI independently predicted MASLD.

Patients with MASLD have a higher mortality rate than the general population, and the primary cause of death in these patients is cardiovascular disease [9,18]. MASLD is often associated with morbidities such as obesity, insulin resistance, T2DM, metabolic syndrome (MetS), hyperlipidemia (dyslipidemia), and cardiovascular diseases (CVD) [12,19].

The prevalence of MASLD is higher in the group with metabolic diseases than in the general population, and MASLD is detected in 56–70% of the patients with T2DM [19,20]. In a systemic review by Lonardo et al., the incidence rate of MASLD in individuals with T2DM was found to range from 50% to 75%, varying according to ethnicity among the studies [21]. In a recent meta-analysis of 156 studies on the association between MASLD and T2DM, MASLD was detected in 65.04% of individuals with T2DM [22]. In a meta-analysis that included 17 studies, the overall prevalence of MASLD was found to be 54%, including 10,897 T2DM patients [23]. Consistent with the findings in the literature, our findings underscore the substantial prevalence of MASLD in patients recently diagnosed with T2DM. The observed rate of 82.4% aligns with the increasing recognition of MASLD as a common comorbidity in the diabetic population.

In a study involving 203 patients with T2DM, MASLD was detected in 71.9% of the patients, and multivariate analysis identified dyslipidemia, elevated LDL, HbA1c, and diastolic blood pressure as independent predictors [24]. In a study involving 874 patients with T2DM, sex, age, total cholesterol level, BMI, waist circumference, diastolic blood pressure, serum uric acid level, duration of illness, and HDL cholesterol level were identified as independent predictors of the development of MASLD [25]. The scores used to determine the degree of fibrosis in MASLD patients include the AST/platelet ratio index, fibrosis index-based 4, and MASLD fibrosis score. Age; BMI; DM status; AST, ALT, and albumin levels; and platelet count are used when calculating these scoring systems [26,27].

Our correlation analysis revealed significant associations between MASLD of various grades and key metabolic parameters. Waist circumference, LDL cholesterol, and platelet count exhibited positive correlations with MASLD grades. Elevated ALT level was identified as an independent predictor of MASLD in regression analysis, underscoring the liver-centric nature of MASLD. Additionally, BMI emerged as another independent predictor, emphasizing the contribution of overall adiposity to the development and progression of MASLD. Similar to scoring systems, the fact that ALT and BMI were independent predictors in our study shows that our results are compatible with the findings in the literature. The significant correlation of MASLD with waist circumference and LDL suggests that these factors can be added to scoring systems.

The pathophysiological link between T2DM and MASLD involves several interconnected mechanisms. Hepatic fat accumulation disrupts normal energy metabolism and triggers inflammatory signaling pathways that contribute to insulin resistance. In individuals with MASLD, increased secretion of diabetogenic hepatokines such as retinol-binding protein 4 (RBP4), fetuin-A, and fibroblast growth factor 21 (FGF-21) has been observed. Additionally, elevated levels of inflammatory biomarkers—including C-reactive protein, tumor necrosis factor-alpha (TNF-α), and interleukin-6 (IL-6)—along with enhanced hepatic gluconeogenesis and glycogen synthesis, further exacerbate metabolic dysregulation. These processes collectively increase the risk of developing or worsening T2DM [12,28,29].

Chronic hyperinsulinemia due to insulin resistance increases hepatic fat accumulation and the release of triglyceride-rich lipoproteins; this condition results in increased insulin resistance in the liver and occurrence of atherogenic dyslipidemia (combination of LDL cholesterol and hypertriglyceridemia), which are the main factors for increased cardiovascular risk in patients with T2DM. Mitochondrial dysfunction, cytokines, lipotoxins, and adipocytokines play an important role in both MASLD and T2DM [30]. However, insulin resistance in adipocytes causes lipolysis and lipotoxicity via the excessive release of free fatty acids and glycerol into the bloodstream, resulting in impaired insulin secretion through excessive lipid intake in tissues such as the liver, pancreas, and muscles [31]. Free fatty acids promote the production of reactive oxygen species in the liver and activate a fibrogenic response, causing the progression to MASLD and cirrhosis. T2DM and MASLD affect each other’s course as well as increase the risk of development of the other via a common pathophysiological mechanism [32]. While the presence of T2DM can cause progression to MASLD or fibrosis, MASLD also leads to worsening of the natural course of diabetic complications (both micro and macrovascular) in T2DM patients [33]. A key strength of this study is its prospective design and focus on newly diagnosed T2DM patients, which provides early insight into MASLD risk in a real-world clinical setting. Additionally, the study reflects practical and accessible diagnostic methods commonly used in primary care.

This study has several limitations. First, the sample size was not determined based on a prior power calculation, and the relatively small number of patients, particularly in the non-MASLD group, may have limited the ability to detect statistically significant differences in metabolic parameters such as obesity, lipid levels, and hypertension. The study was conducted in a single-center, ethnically homogenous population, which may limit generalizability. Second, although patient selection criteria have been clarified with a flow chart, selection bias may still exist. Third, the cross-sectional design prevents causal inferences. Fourth, conventional ultrasonography is operator dependent and lacks the sensitivity and specificity required for accurately diagnosing MASLD; the absence of elastographic assessment limits the evaluation of disease severity. Lastly, non-invasive scoring systems such as the Fatty Liver Index (FLI), Hepatic Steatosis Index (HSI), or the Hepamet Score were not used, which could have enhanced diagnostic precision and clinical applicability. Advanced diagnostic methods, such as liver biopsy or advanced imaging techniques, may increase the accuracy of MASLD diagnosis. Additionally, parameters such as uric acid, blood pressure, and waist-to-height ratio were not collected. Advanced imaging like elastography or MRI was not used due to institutional limitations and cost concerns.

## 5. Conclusions

In conclusion, our investigation contributes to the growing body of literature elucidating the intricate link between MASLD and newly diagnosed T2DM. While several larger-scale studies exist, our study provides a distinct contribution by specifically targeting newly diagnosed T2DM patients—an underrepresented and clinically critical group. The implications of our findings extend to the clinical realm. The high prevalence of MASLD in this population emphasizes the need for routine screening and vigilant management. Further research should delve into mechanistic insights and explore tailored therapeutic strategies for this high-risk population.

## Figures and Tables

**Figure 1 jcm-14-03513-f001:**
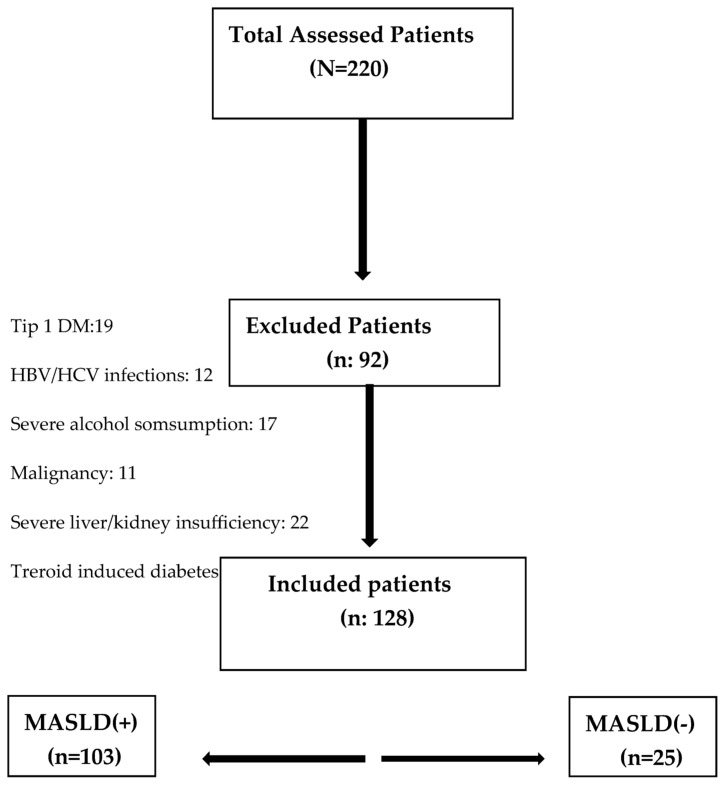
Flowchart illustrating patient selection, exclusion criteria, and group classification based on the presence or absence of MASLD.

**Figure 2 jcm-14-03513-f002:**
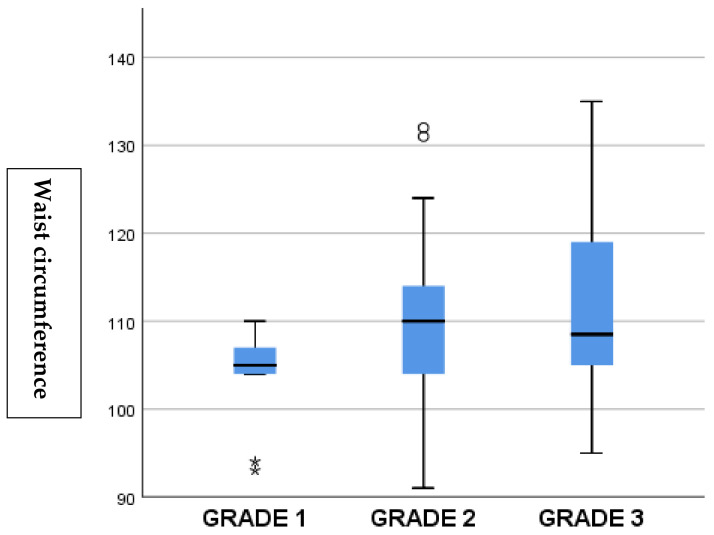
Association between MASLD grades and waist circumference. Higher grades of steatosis were associated with increased waist circumference, reflecting greater central obesity.

**Figure 3 jcm-14-03513-f003:**
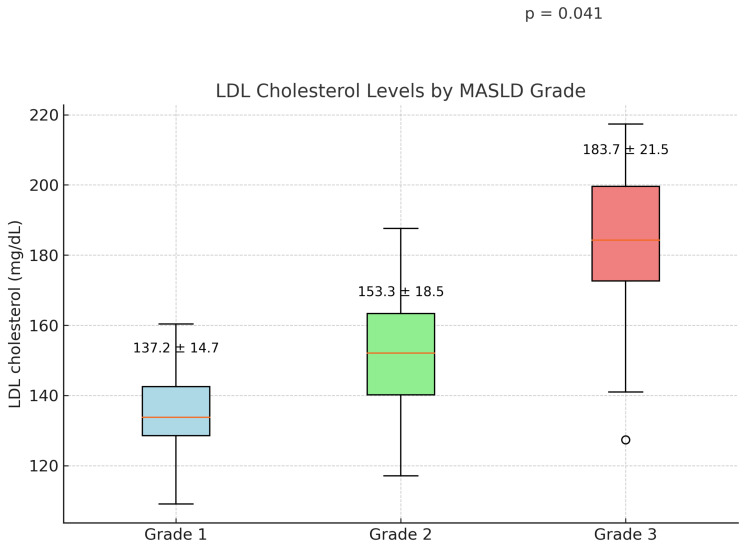
Relationship between MASLD grades and LDL cholesterol levels. A stepwise increase in LDL levels was observed with increasing steatosis severity, underlining the link between hepatic fat accumulation and dyslipidemia.

**Table 1 jcm-14-03513-t001:** Clinical and demographic characteristics of patients with and without hepaticsteatosis.

Without MASLD		With MASLD	*p* Value
(n = 25)		(n = 103)	
Age (years)	57.8	54.9	
Sex, female	11 (34.3)	39 (40.6)	0.631
BMI (kg/m^2^)	30.6 ± 4.2	33.4 ± 5.3	0.130
Waist circumference (cm)	109.9 ± 7.8	109.7 ± 9.6	0.884
Retinopathy, n (%)	7 (21.8)	27 (28.1)	0.753
Nephropathy, n (%)	8 (25)	31 (32.2)	0.530
Neuropathy, n (%)	5 (15.6)	23 (23.9)	0.355
CAD, n (%)	4 (12.5)	32 (33.3)	0.128
Hypertension, n (%)	16 (50.3)	55 (57.2)	0.648
Smoking, n (%)	11 (34.3)	35 (34.4)	0.650
cIMT (mm)	0.72 ± 0.21	0.81 ± 0.27	0.259
Glucose (mg/dL)	163.7 (92–437)	167.6 (100–630)	0.381
HbA1c (%)	8.6 ± 2.5	9.2 ± 3.7	0.341
LDL cholesterol (mg/dL)	145.8 ± 33.5	155.9 ± 49.8	0.457
Triglyceride (mg/dL)	170.6 (58–224)	205.8 (58–1314)	0.057
Non-HDL cholesterol (mg/dL)	167.9 ± 28.7	189.1 ± 57.5	0.048
HDL cholesterol (mg/dL)	47.8 ± 10.8	44.2 ± 10.6	0.492
AST (U/L)	23.42 (11–51)	24.8 (8–176)	0.165
ALT (U/L)	24.9 ± 8	47.1 ± 23	<0.001
Creatinine (mg/dL)	0.87 (0.5–1.0)	0.81 (0.5–6)	0.091
Platelet count (10^3^/μL)	246 ± 56	252 ± 62	0.672

Abbreviations: MASLD, Metabolic Dysfunction-Associated Steatotic Liver Disease; BMI index, Body mass index; CAD, coronary artery disease; cIMT, Carotid intima-media thickness; HbA1c, Glycosylated hemoglobin; LDL, low-density lipoprotein; HDL, high-density lipoprotein; AST, aspartate transferase; ALT, alanine transaminase.

**Table 2 jcm-14-03513-t002:** Relationship of fatty liver grades with other variables.

	r	*p*-Value *
cIMT	−0.017	0.867
BMI	0.216	0.077
Waist circumference	0.234	0.032
Glucose	−0.030	0.788
HbA1c	0.102	0.346
LDL cholesterol	0.263	0.021
Triglyceride	0.142	0.222
Non-HDL	0.226	0.046
AST	0.182	0.121
ALT	0.187	0.119
Creatinine	0.112	0.354
Platelet count	0.226	0.040
HDL cholesterol	−0.039	0.547

*: Spearman’s Correlation. Abbreviations: cIMT, Carotid intima-media thickness; BMI, body mass index; HbA1c, Glycosylated hemoglobin; LDL, low-density lipoprotein; HDL, high-density lipoprotein; AST, aspartate transferase; ALT, alanine transaminase.

**Table 3 jcm-14-03513-t003:** Independent factors affecting MASLD in T2DM patients in stepwise multiple linear regression analysis.

Coefficients ^a^				
Model	Unstandardized Coefficients		Standardized Coefficients	*p*-Value
	B	Std. Error	Beta	
(Constant)	−0.354	0.660		
ALT	0.00	0.002	0.332	0.001
BMI	0.051	0.021	0.256	0.013
**Excluded Variables ^b^**				
**Model**	**B**	**Partial Correlation**	**Collinearity** **Statistics**	***p***-**Value**
			**Tolerance**	
cIMT	0.051	0.055	0.974	0.611
Waist circumference	−0.128	−0.093	0.465	0.383
Glucose	0.081	0.097	0.988	0.371
HbA1c	0.169	0.191	0.972	0.080
LDL cholesterol	0.091	0.095	0.914	0.389
Triglyceride	0.149	0.164	0.993	0.136
Non-HDL	0.142	0.152	0.924	0.160
AST	−0.259	−0.120	0.173	0.269
Creatinine	0.159	0.162	0.976	0.140
Platelet count	0.104	0.114	0.957	0.321
HDL cholesterol	−0.056	−0.051	0.982	0.601

^a^ Dependent variable: MASLD. ^b^ Independent variables: (Constant), ALT, BMI. Abbreviations: MASLD, Metabolic Dysfunction-Associated Steatotic Liver Disease; T2DM, type 2 diabetes mellitus; cIMT, carotid intima-media thickness; BMI, body mass index; HbA1c, glycosylated hemoglobin; LDL, low-density lipoprotein; HDL, high-density lipoprotein; AST, aspartate transferase; ALT, alanine transaminase.

## Data Availability

All of our authors agree on the sharing of research data.

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
