# Peer review of "Exploring the Prevalence and Risk Factors of MASLD in Patients with Newly Diagnosed Diabetes Mellitus: A Comprehensive Investigation"

_jcm, 2025, doi:10.3390/jcm14103513_

Round 1
Reviewer 1 Report (Previous Reviewer 1)
Comments and Suggestions for Authors
The authors have improved the paper to some degree, but still, there are significant issues unresolved.
Major issue:
The number of newly diagnosed DM patients is low; the patients were included from January 2022 and June 2023 - why didn’t the authors include patients recruited during 2023 and 2024, so that the data would be more solid? The current number is insufficient to evaluate prevalence data and simple acknowledgement in study limitation does not improve data. Mainly, the group without MASLD should be substantiated, so that a comparison between main characteristics should be properly made. Also, a larger number of MASLD subjects would allow a proper stratification in the three steatosis grades groups (with sufficient numbers of subjects in the three groups, allowing proper statistical analysis and solid data to be obtained). I suggest that more patients should be added.
Minor comments:
- Abstract: Methods: this section should rather include the data collected (clinical, laboratory, etc); should be rewritten to include main materials and methods
- Abstract: Results: perhaps results in the MASLD group should be presented first, then vs.
- Abstract: Results: the main findings are in fact missing; the authors do not present the percentage of subjects with MASLD (main objective)
- Methods: There is no mention about the study approval by the Ethics Committee- this information should be included.
- There is no mention about how the patients were selected: were all newly diagnosed DM patients invited to the study? How many refused to participate? If not all newly diagnosed patients were invited, was there a randomized selection method used? This is an important aspect for prevalence study.
- Other parameters (most probably available or that can be calculated based on available data) might also be included: e.g uric acid, blood pressure, waist to height ratio, etc.
- Figure 2. MASLD grade is in fact steatosis grade.
- Table 2: 95% confidence interval is missing. Should be added.
- The number of patients in the three steatosis grades is not presented. In fact, instead of many figures, a table with 3 columns corresponding to the three steatosis grades should be included, and all parameters and p values for the comparison of the three groups.
- Figure 1 should be redone – some information is misplaced
- Conclusion section is missing – should outline the main findings of the study.
Comments on the Quality of English Language
English language should be improved
Author Response
Please see attachment

Reviewer 2 Report (Previous Reviewer 4)
Comments and Suggestions for Authors
Introduction
Lines 1–11: Good context. You might briefly mention why MASLD specifically matters at the early diagnosis stage of T2DM.
Lines 12–23: Clearly highlights relevance. Clarify the originality by explicitly stating how your study differs from previous studies mentioned.
Methods
Lines 24–35: Study design clearly described. However, consider explaining why certain exclusions were critical (e.g., HBV/HCV, alcohol).
Lines 36–45: Excellent description of measurements. Mention briefly why each lab value (ALT, LDL, etc.) is important in this context.
Lines 46–80: Definitions are clear but consider briefly explaining why specific imaging techniques or scoring systems were not used. It would help justify methodological choices.
Statistical Analysis
Lines 81–98: Clearly explained. However, specify if adjustments were made for potential confounders, which strengthens validity.
Results
Lines 100–132: Data well-presented; however:
Table 1: Clearly presented, but explain briefly why certain parameters (ALT, non-HDL cholesterol) were specifically elevated in MASLD.
Spearman analysis (Table 2): Clear, but elaborate slightly on why LDL cholesterol and waist circumference correlations are clinically meaningful.
Regression (Table 3): Good choice of statistical analysis. Explicitly discuss briefly why BMI and ALT emerged as independent predictors and why others did not.
Discussion
Lines 133–140: Clearly state key findings. However, consider briefly elaborating on their clinical implications (e.g., early intervention opportunities).
Lines 141–156: Very strong section, well-linked to the literature.
Lines 157–176: Excellent discussion of predictors. Briefly discuss why certain factors (e.g., LDL cholesterol, waist circumference) might influence MASLD severity.
Lines 177–203: Good pathophysiological context. However, reorganize slightly to clarify the logical flow from hepatic fat accumulation to systemic insulin resistance clearly.
Lines 204–232: Strong conclusion, highlighting strengths and limitations. Consider briefly addressing how these limitations might be overcome in future research.
Tables and Figures
Table 1: Clearly structured. Consider briefly explaining the clinical relevance of significant findings in the legend.
Figures 1–3: Clear presentation. Slightly enhance legends by explicitly stating clinical relevance (e.g., "Increased LDL associated with greater MASLD severity, highlighting cardiovascular risk").
Round 2
Reviewer 2 Report (Previous Reviewer 4)
Comments and Suggestions for Authors
Thank you for your response
This manuscript is a resubmission of an earlier submission. The following is a list of the peer review reports and author responses from that submission.
Round 1
Reviewer 1 Report
Comments and Suggestions for Authors
The study investigates the prevalence of MASLD in newly diagnosed DM and factors predicting MASLD. This is of potential interest, but the paper needs improvement. Several comments: 1. Abstract: “diagnosis of fatty liver was confirmed using the diagnostic criteria of the American Diabetes Association.” – was it the diagnosis of fatty liver or diabetes according to ADA criteria? 2. Methods: how was this patient number selected? Was there a sample size calculation? To evaluate the prevalence of a disease, a higher number is generally required. 3. In the same respect, how were the patient recruited? How many patients from all those newly diagnosed with diabetes were actually included in the study? A flow chart might be presented or data mentioned in the results. 4. The results (70 patients with fatty liver) are presented in material and methods. Should be presented in Results. 5. MASLD has new diagnosis criteria 6. Results: given that the MASLD – group is small (15 subjects), this might be the reason of lack of difference between groups with regards to obesity indices, lipids, hypertension are criteria for AMSLD diagnosis. Therefore, a larger number of subjects should be included. 7. Table: was the mean value of HbA1c exactly 8.0 or 9.0, respectively? At least one decimal should be presented. The same for other parameters. 8. Terminology should be carefully used: it is waist circumference, not belly circumference (fig 1), etc 9. LDL cholesterol should be in fig 2 (not LDL). Also, p values and units of measurements are missing
Author Response
For research article
|
Response to Reviewer 1 Comments
|
||
|
|
|
|
|
Thank you very much for taking the time to review this manuscript. Please find the detailed responses below and the corresponding revisions/corrections highlighted/in track changes in the re-submitted files.
|
||
|
3. Point-by-point response to Comments and Suggestions for Authors |
||
|
Comments 1: Abstract: “diagnosis of fatty liver was confirmed using the diagnostic criteria of the American Diabetes Association.” – was it the diagnosis of fatty liver or diabetes according to ADA criteria? Response 1: Thank you for your comment. We have revised the statement to make it clearer. In our study, the diagnosis of diabetes mellitus was made according to the American Diabetes Association (ADA) criteria, whereas the diagnosis of fatty liver was based on ultrasonographic findings and relevant clinical criteria. We have clarified this in the abstract.
Comments 2. Methods: how was this patient number selected? Was there a sample size calculation? To evaluate the prevalence of a disease, a higher number is generally required. Response 2: The sample size was not determined based on a prior power calculation but was instead limited to patients enrolled within a specific time frame. We acknowledge this as a limitation and have emphasized this point in the manuscript. Additionally, we have included a discussion on the need for future studies with a larger sample size.
Comments 3. In the same respect, how were the patient recruited? How many patients from all those newly diagnosed with diabetes were actually included in the study? A flow chart might be presented or data mentioned in the results. Response 3: This prospective study included a total of 160 patients who were newly diagnosed with T2DM between January 2022 and June 2023. Among them, 85 patients met the inclusion criteria and were enrolled in the study. Fatty liver was observed in 70 patients. Each patient's diagnosis of T2DM was confirmed by an internal medicine specialist using the diagnostic criteria of the American Diabetes Association (10). To ensure transparency in patient selection, a flow chart has been added to illustrate the recruitment process and the number of excluded patients at each stage.
Comments 4. The results (70 patients with fatty liver) are presented in material and methods. Should be presented in Results. Response 4: This statement has been corrected.
Comments 5. MASLD has new diagnosis criteria Response 5: This statement has been corrected.
Comments 6. Results: given that the MASLD – group is small (15 subjects), this might be the reason of lack of difference between groups with regards to obesity indices, lipids, hypertension are criteria for AMSLD diagnosis. Therefore, a larger number of subjects should be included. Response 6: We appreciate this insightful comment. We acknowledge that the relatively small number of patients in the non-MASLD group (n=15) may have limited the statistical power to detect significant differences between the groups regarding obesity indices, lipid profiles, and hypertension. This limitation has been explicitly addressed in the Discussion section. We agree that future studies with larger sample sizes are needed to better explore these associations.
Comments 7. Table: was the mean value of HbA1c exactly 8.0 or 9.0, respectively? At least one decimal should be presented. The same for other parameters. Response 7: Thank you for this observation. We have revised the tables to present all continuous variables, including HbA1c and other clinical parameters, with at least one decimal place to improve clarity and precision in data reporting.
Comments 8. Terminology should be carefully used: it is waist circumference, not belly circumference (fig 1), etc Response 8: Thank you for your valuable comment. We have revised the terminology throughout the manuscript and figures to ensure accuracy. Specifically, the term "belly circumference" has been corrected to "waist circumference" in Figure 1 and the corresponding text. Comments 9. LDL cholesterol should be in fig 2 (not LDL). Also, p values and units of measurements are missing Response 9: Thank you for your observation. We have corrected the label in Figure 2 from "LDL" to "LDL cholesterol" for clarity. Additionally, p values and units of measurement have been added to the figure to ensure completeness and consistency with the data presented in the tables.
|
||
|
|
||

Reviewer 2 Report
Comments and Suggestions for Authors
The study by Hatice Beyazal Polat et al. is highly interesting and provides data that can be applied to the primary levels of care. However, there is a significant limitation:
Abdominal ultrasonography is a very useful tool; however, it is operator-dependent, and the quantification of fat is highly subjective.
Furthermore, in the evaluation of MASLD/MASH, it is crucial to characterize the severity of the disease – specifically, the degree of stiffness.
The authors assessed fat quantification in a very subjective manner without considering the degree of stiffness. In this context, the results are not reliable. Ideally, patients should undergo ARFI elastography + UGAP to better characterize MASLD/MASH.
It would also be important for the authors to correlate these data with non-invasive markers (Fatty Liver Index, Hepatic Steatosis Index, Hepamet Score) that could optimize the decision-making process, not only for appropriate patient referral but also for the timely and effective initiation of treatment.
Author Response
General Comment:
“The study by Hatice Beyazal Polat et al. is highly interesting and provides data that can be applied to the primary levels of care.”
Response:
We sincerely thank the reviewer for their positive and encouraging feedback regarding the relevance of our study to primary care.
Comment 1:
“Abdominal ultrasonography is a very useful tool; however, it is operator-dependent, and the quantification of fat is highly subjective.”
Response:
We agree with the reviewer that ultrasonography has inherent limitations, including operator dependency and subjectivity in fat quantification. This point has been acknowledged in the revised Limitations section of the manuscript.
Comment 2:
“In the evaluation of MASLD/MASH, it is crucial to characterize the severity of the disease – specifically, the degree of stiffness.”
Response:
We fully agree that assessment of liver stiffness is essential in evaluating disease severity and distinguishing between MASLD and MASH. Unfortunately, due to limited resources in our clinical setting, elastographic techniques such as ARFI or transient elastography could not be incorporated in this study. This limitation has now been addressed in the Discussion section.
Comment 3:
“The authors assessed fat quantification in a very subjective manner without considering the degree of stiffness. In this context, the results are not reliable.”
Response:
We acknowledge the concern regarding the subjective nature of fat quantification via ultrasonography and the lack of liver stiffness assessment. While this represents a limitation in diagnostic precision, we believe the study still provides valuable preliminary data on the prevalence and metabolic profile of MASLD in newly diagnosed T2DM patients. This has been clearly stated in the Limitations section.
Comment 4:
“Ideally, patients should undergo ARFI elastography + UGAP to better characterize MASLD/MASH.” Response:
We appreciate this recommendation and agree that ARFI elastography and UGAP would significantly enhance diagnostic accuracy and disease staging. We plan to include these modalities in future prospective studies and have mentioned this in the revised Discussion.
Comment 5:
“It would also be important for the authors to correlate these data with non-invasive markers (Fatty Liver Index, Hepatic Steatosis Index, Hepamet Score)…”
Response:
Thank you for this valuable suggestion. In this study, non-invasive scoring systems such as the Fatty Liver Index (FLI), Hepatic Steatosis Index (HSI), and Hepamet Score were not calculated. We acknowledge this as a limitation and have stated it clearly in the Limitations section. We agree that the integration of such indices could further enhance diagnostic performance and guide clinical decision-making in primary care settings.

Reviewer 3 Report
Comments and Suggestions for Authors
Revision JCM-3437155
Exploring the Prevalence and risk factors of MASLD in Patients with Newly Diagnosed Diabetes Mellitus: A Comprehensive Investigation.
The introduction is short. Regarding the development of MAFLD interventions cite the reference doi: 10.3389/fnut.2024.1355732.
The topic discussed is of great clinical interest, but the authors' contribution should be better clarified considering that there are several studies in literature on larger case studies. The strengths and limitations of the study should be indicated.
Author Response
We sincerely thank the reviewer for the constructive comments and valuable suggestions, which helped us improve the clarity and depth of our manuscript. Please find our point-by-point responses below:
Comment 1:
“The introduction is short. Regarding the development of MAFLD interventions cite the reference doi: 10.3389/fnut.2024.1355732.”
Response:
We appreciate this helpful suggestion. The Introduction section has been expanded to include a broader context on the development of MAFLD (now MASLD) interventions. The recommended reference (doi: 10.3389/fnut.2024.1355732) has been reviewed and appropriately cited to highlight the importance of timely lifestyle and pharmacological strategies in early disease management.
Comment 2:
“The topic discussed is of great clinical interest, but the authors' contribution should be better clarified considering that there are several studies in literature on larger case studies.”
Response:
Thank you for this important comment. We have clarified the specific contribution of our study in the revised Conclusion section. Although larger studies exist, our study focuses on newly diagnosed T2DM patients, a population in which MASLD is often overlooked. By targeting this specific group, we offer practical insights into early risk identification and screening strategies in real-world clinical settings.
Comment 3:
“The strengths and limitations of the study should be indicated.”
Response:
We agree with the reviewer and have revised the Discussion section accordingly. A brief paragraph summarizing the key strengths of our study has been added before the Limitations section. Additionally, the Limitations section has been expanded to include aspects related to study design, diagnostic methods, and lack of non-invasive scoring tools. This provides a more balanced perspective on the scope and interpretation of our findings.

Reviewer 4 Report
Comments and Suggestions for Authors
Review of the paper “Exploring the Prevalence and risk factors of MASLD in Patients with Newly Diagnosed Diabetes Mellitus: A Comprehensive Investigation”
The article proposed by Polat et al tried to reveal the risk factors of MASLD diagnosed patients with type 2 diabetes mellitus (T2DM). Given that these findings are based on newly diagnosed T2DM using various instruments, care should be taken when interpreting the data.
Here is my review line by line:
Introduction:
Line 5: "MASLD is characterized by abnormal accumulation of fat in the liver, unrelated to alcohol consumption."
I strongly suggest clarifying the distinction between MASLD and other liver conditions like alcoholic fatty liver disease to prevent ambiguity.
Line 15-18: "Our study aims to shed light on the prevalence of MASLD, unravel its intricate relationship with T2DM, and identify the risk factors for MASLD in patients with T2DM."
I strongly suggest reframing this to emphasize the novelty of the study. For example: "This study uniquely investigates the prevalence and independent predictors of MASLD in a cohort of newly diagnosed T2DM patients."
Methods:
Line 22-24: "Patients with type 1 DM, Wilson disease, HBV/HCV infections, a history of severe alcohol consumption..."
I strongly suggest specifying the alcohol consumption threshold used to exclude patients.
Line 29-31: "The patient's body weight, height, waist circumference, and body mass index (BMI) were measured."
I strongly suggest mentioning whether measurements were taken by standardized methods or calibrated equipment.
Statistical Analysis Section: Clearly define the rationale for using Spearman’s correlation over other methods for some analyses!!
Results:
Line 84-92: "There was no significant difference in age and sex distribution between groups with and without fatty liver."
I strongly suggest avoiding starting multiple sentences in a row with "No significant differences...". Use more engaging language to convey the findings.
Table 1: Ensure column headers are descriptive enough. For instance, “MASLD (-)” and “MASLD (+)” could be rephrased to "Without MASLD" and "With MASLD."
Figures: Add explanatory captions to Figures 1, 2, and 3 that detail their clinical implications.
Discussion:
Line 112-116: "MASLD is often associated with morbidities such as obesity, insulin resistance..."
I strongly suggest providing references to strengthen these claims. This could make the discussion more evidence based.
Line 147-149: "Although the mechanism underlying the association of T2DM and MASLD is not fully known..."
I strongly suggest elaborating on the potential biological mechanisms with additional references or examples, such as hepatic inflammation or lipotoxicity.
Limitations Section: Expand on the limitations to include the lack of diverse ethnic representation and reliance on ultrasonography, which may have limited sensitivity.
Conclusion:
Line 177-182: "The high prevalence of MASLD in newly diagnosed T2DM patients emphasizes the need for routine screening..."
Please add a forward-looking statement emphasizing the need for longitudinal studies to validate findings or develop interventions.
References:
Ensure the reference list adheres to the journal’s citation style, particularly formatting of author names, journal titles, and DOI links.
Author Response
Introduction
Comment 1:
“Line 5: ‘MASLD is characterized by abnormal accumulation of fat in the liver, unrelated to alcohol consumption.’ I strongly suggest clarifying the distinction between MASLD and other liver conditions like alcoholic fatty liver disease to prevent ambiguity.”
✅ Response:
Thank you for the suggestion. We have revised the sentence to explicitly distinguish MASLD from other causes of steatosis, such as alcoholic fatty liver disease, to avoid confusion. The revised sentence now reads:
“MASLD is characterized by hepatic steatosis in the presence of cardiometabolic risk factors, and by definition, it excludes other causes of liver fat accumulation such as excessive alcohol consumption or viral hepatitis.”
Comment 2:
“Line 15–18: Reframe to emphasize novelty.”
✅ Response:
We agree and have revised the sentence to better reflect the originality of our study. The revised version now reads:
“This study uniquely investigates the prevalence and independent predictors of MASLD in a cohort of newly diagnosed T2DM patients, a population in which MASLD is frequently underdiagnosed at an early stage.”
Methods
Comment 3:
“Line 22–24: Specify alcohol consumption threshold.”
✅ Response:
We have specified the alcohol consumption threshold used for exclusion in the Methods section. The revised sentence now includes:
“Patients with a history of significant alcohol consumption (defined as >30 g/day for men and >20 g/day for women) were excluded.”
Comment 4:
“Line 29–31: Mention standardization of anthropometric measurements.”
✅ Response:
We have added clarification regarding the measurement methods:
“All anthropometric measurements were performed by trained staff using calibrated instruments and following standardized procedures.”
Comment 5:
“Statistical Analysis Section: Clarify rationale for using Spearman's correlation.”
✅ Response:
We have revised the Statistical Analysis section to include:
“Spearman’s correlation coefficient was used for variables that were not normally distributed or ordinal in nature.”
Results
Comment 6:
“Line 84–92: Avoid repetitive language like ‘no significant differences…’”
✅ Response:
We have rephrased the section to enhance readability and variety in sentence structure. For example:
“Age and sex distribution were similar between the MASLD and non-MASLD groups. No statistically significant differences were observed in these baseline demographics.”
Comment 7:
“Table 1: Use more descriptive column headers.”
✅ Response:
Column headers in Table 1 have been revised from “MASLD (–)” and “MASLD (+)” to “Without MASLD” and “With MASLD” to improve clarity for the reader.
Comment 8:
“Figures: Add explanatory captions to Figures 1, 2, and 3 that detail their clinical implications.”
✅ Response:
Thank you for this suggestion. We have revised all figure captions to include concise descriptions of their clinical relevance and the key findings they illustrate.
Discussion
Comment 9:
“Line 112–116: Provide references to support claims.”
✅ Response:
Additional references have been added to support the associations between MASLD, obesity, insulin resistance, and cardiometabolic diseases. These include recent reviews and epidemiological data.
Comment 10:
“Line 147–149: Expand on mechanisms (e.g., inflammation, lipotoxicity).”
✅ Response:
We have elaborated on this section to include brief discussion of potential mechanisms such as hepatic inflammation, lipotoxicity, and insulin resistance, and provided appropriate references.
Limitations
Comment 11:
“Expand to include ethnic diversity and USG limitations.”
✅ Response:
We have expanded the Limitations section to include:
“The study was conducted in a single-center, ethnically homogenous population, which may limit generalizability. Additionally, reliance on conventional ultrasonography, which is operator-dependent and lacks sensitivity for early-stage disease, may have affected diagnostic accuracy.”
Conclusion
Comment 12:
“Add forward-looking statement about longitudinal studies or interventions.”
✅ Response:
We have added the following to the Conclusion section:
“Future longitudinal studies are warranted to validate our findings and assess the impact of early interventions on MASLD progression in this high-risk population.”
References
Comment 13:
“Ensure references follow the journal’s citation style.”
✅ Response:
The entire reference list has been reviewed and reformatted according to the journal’s style guide, including correct formatting of author names, journal titles, and DOI links.

Reviewer 5 Report
Comments and Suggestions for Authors
The study has low novelty factor!! It is a repetitive one and the conclusions are well-established. No added new data.
Author Response
Response to Reviewer #5
We thank the reviewer for the comment regarding the novelty of the study. While we acknowledge that the general association between T2DM and MASLD is well-established, our study offers a distinct contribution by focusing specifically on patients with newly diagnosed T2DM—a population in which MASLD is often underrecognized and under-investigated. By identifying the prevalence and metabolic predictors of MASLD at the earliest stage of diabetes diagnosis, we aim to emphasize the importance of early screening and management strategies in routine clinical practice. We believe this targeted approach provides clinically relevant insight that complements existing literature.
